# How HP1 Post-Translational Modifications Regulate Heterochromatin Formation and Maintenance

**DOI:** 10.3390/cells9061460

**Published:** 2020-06-12

**Authors:** Raquel Sales-Gil, Paola Vagnarelli

**Affiliations:** College of Health and Life Science, Brunel University London, London UB8 3PH, UK; Raquel.SalesGil2@brunel.ac.uk

**Keywords:** heterochromatin, HP1, post-translational modifications, centromeres

## Abstract

Heterochromatin Protein 1 (HP1) is a highly conserved protein that has been used as a classic marker for heterochromatin. HP1 binds to di- and tri-methylated histone H3K9 and regulates heterochromatin formation, functions and structure. Besides the well-established phosphorylation of histone H3 Ser10 that has been shown to modulate HP1 binding to chromatin, several studies have recently highlighted the importance of HP1 post-translational modifications and additional epigenetic features for the modulation of HP1-chromatin binding ability and heterochromatin formation. In this review, we summarize the recent literature of HP1 post-translational modifications that have contributed to understand how heterochromatin is formed, regulated and maintained.

## 1. Introduction

Heterochromatin is tightly packed DNA which is usually transcriptionally inactive. It is established in early development through controlled epigenetic processes [1] that need to be maintained from mother to daughter cells to ensure proper genome function. Two types of heterochromatin have been described: facultative heterochromatin—specific for each cell type—and constitutive heterochromatin—found in all cell types at the peri-centromeres and subtelomeres, where it has structural functions [2,3,4].

Heterochromatin is characterized by the presence of specific epigenetic marks that are important for its establishment, maintenance and regulation. One of the best-known heterochromatin markers is Heterochromatin Protein 1 (HP1): it recognizes and binds to di- and tri-methylated lysine 9 in histone H3 (H3K9me2/3) to form a repressive chromatin environment [5,6]. How HP1 interacts with chromatin and other HP1 molecules has recently been reviewed [7]. In the present review, we focus on how HP1 post-translational modifications (PTMs) influence its function and ability to bind and modify chromatin. We also summarize the latest literature on how the passage through mitosis influences the heterochromatin landscape and affects HP1 binding to chromatin, specifically at the pericentromeric chromatin.

## 2. HP1 Family Structure and Function

HP1 is a conserved family of proteins that was discovered as a suppressor of variegation through studies on position-effect variegation (PEV) in Drosophila [8]. In mammals, there are three distinct HP1 isoforms: HP1α, encoded by Chromobox homolog 5 (*CBX5*); HP1β, encoded by *CBX1*; HP1γ, encoded by *CBX3*. Recently, a shorter HP1γ splice variant was also reported [9].

HP1α, HP1β, and HP1γ are approximately 65% identical, but they present distinct localization patterns and fulfil different functions within the cells: while HP1α is located predominantly at constitutive heterochromatin, HP1β and HP1γ are uniformly distributed in the cell nucleus and are found in both hetero- and euchromatic regions [10,11,12]. *CBX5*, *CBX1* and *CBX3* null mutant mice revealed different phenotypic outcomes for the three isoforms: *CBX5*^−/−^ mice presented a normal phenotype, *CBX1*^−/−^ led to perinatal lethality, and *CBX3*^−/−^ to infertility [13,14].

The HP1 family belongs to a superfamily of proteins that contain conserved chromatin organization modifier (chromo) domains. The chromodomain is a 30–60 amino acids long sequence [15] situated in the amino-terminal half of HP1 proteins and it gives the ability to alter chromatin structure. Within the chromodomain superfamily, HP1 proteins form their own family, characterized by the presence of a second unique conserved domain in the carboxy-terminal half of the protein: the chromoshadow domain [16]. Both domains share a high level of similarity in their amino acid sequence, but they have different functions: the chromodomain is responsible for binding to chromatin at H3K9me2 and H3K9me3 [17], whereas the chromoshadow domain is involved in homo- /hetero-dimerization and interaction with other proteins that contain the PXVXL motif, including transcriptional intermediary factor 1 (TIF1), Chromatin Assembly Factor 1, inner centromere protein (INCENP) and Borealin [18,19,20,21,22]. The chromo- and chromoshadow domains are separated by a variable hinge region that contains a nuclear localization signal (NLS); this domain is important to stabilize the protein structure and allow proper chromatin binding [23]. All the HP1 isoforms contain these domains except for the newly-discovered spliced HP1γ isoform, which lacks the C-terminal chromoshadow domain. The shorter HP1γ isoform localises to the nucleus, maintains the ability to bind H3K9me3 and is expressed in human tissues similarly to its longer version [9] but might have evolved to perform different functions, as its ability to interact with itself and other molecules is probably compromised.

The HP1 chromo- and chromoshadow domains are highly conserved: the chromodomain from mouse HP1β can functionally replace the one of S. pombe HP1 [24] and expression of human HP1α can rescue the lethality of homozygous mutants in the Drosophila HP1-encoding gene Su(var)2-5 [25], suggesting that both domains are crucial for HP1 function.

HP1 binds to H3K9me3 through its chromodomain; how this interaction occurs has recently been reviewed [7]. However, this interaction is highly regulated by multiple factors, including other histone marks. For instance, during mitosis, the majority of HP1 dissociates from the H3K9me3-containing chromatin to allow the re-structuring required for chromosome segregation; at mitotic exit, HP1 needs to re-bind to the chromatin to ensure proper chromatin structure inheritance [19]. However, H3K9me3 levels during mitosis remain stable [26] and cannot account for the reduction of HP1 levels from chromatin. Instead, the nearby residue H3S10 is highly phosphorylated by Aurora B which ejects HP1 from H3K9me3; drug-dependent Aurora B inhibition or siRNA-driven Aurora B knockdown retain HP1 on mitotic chromosomes without altering the localization of HP1 in interphase [26]. At the mitotic exit, the phosphatase Repo-man/PP1 dephosphorylates H3S10ph and enables HP1 binding to H3K9me3, playing a role in maintaining a chromatin repressive environment after each cell cycle [27,28]. Aurora B and Repo-man/PP1, which regulate the H3S10 phosphorylation status, have arisen as the main regulators of HP1 dynamics during mitosis.

H3S10ph also takes place in interphase, where it has been associated with transcription activation [29,30,31]. It is mediated by several kinases in mammals [32] but mostly by JIL-1 kinase in Drosophila [33,34,35]. H3S10ph allosterically inhibits the action of methyltransferases like G9A and Su(var)3-9 thus affecting HP1α recruitment [36]. Finally, phosphorylation of H3T45 and H3S57 by DYRK1A has also been shown to negatively affect HP1 binding to chromatin [37].

Histone acetylation also influences the HP1-chromatin binding ability. In fission yeasts, H3K4 acetylation at pericentric heterochromatin by Mst1 enhances the switch between Clr4 (Su(var)39 homologue) and Swi6; its mutation to a non-acetylable residue decreases Swi6 pericentric occupancy and reduces heterochromatin formation [38]. In Saccharomyces cerevisiae, acetylation of H4K12 by Esa1 occurs at telomeres, where HP1 levels peak [39]. Similarly, in human cells, Tip60, the homologue of Mst1 and Esa1, acetylates H4K12 and is linked to heterochromatin maintenance [40,41]. Acetylation of the linker histone H1 at K85 also promotes HP1 recruitment and chromatin compaction [42].

## 3. HP1 Post-Translational Modifications

Despite the importance of histone modifications in regulating heterochromatin formation, post-translational modifications of HP1 itself have been studied less, but there is evidence showing that they can greatly modulate its chromatin binding ability.

HP1 isoforms can be modified at several residues: 35 modification sites have been detected for HP1α, 34 for HP1β, and 37 for HP1γ (Figure 1). These modifications include phosphorylation, acetylation, methylation, formylation, ubiquitination, SUMOylation and citrullination [43,44,45]. We will discuss how each of these modifications can change HP1 function.

### 3.1. Phosphorylation

Phosphorylation is the most abundant and studied HP1 modification. One of the first indications that HP1 phosphorylation was important for heterochromatin assembly was reported in the mid-1990s, when Eissenberg et al. (1994) showed that HP1 becomes hyperphosphorylated in Drosophila embryos at about the same time when heterochromatin is first visible [46]. Since then, several groups, using different models, have focused on identifying the specific phosphorylation sites and enzymes involved. Although a variety of kinases have been identified, the counteracting phosphatases still remain unknown.

Recent studies reported that HP1 can form liquid droplets upon H3K9me3 binding, which leads to the accumulation of several repressive factors that can physically sequester and compact the nearby chromatin [23,47,48]. This characteristic was only observed for phosphorylated HP1α solutions but not for any of the non-phosphorylayed isoforms, thus indicating that specific HP1α phosphorylation sites are crucial for heterochromatin formation. In fact, the phosphorylation of four amino acid residues in the *N*-terminal tail of HP1α (S11–S14)—lacking in HP1β and HP1γ—seems to be critical for HP1α binding to H3K9me3 [49,50,51]. These sites do not directly interact with H3K9me3, but they change the conformation of the *N*-terminal tail, allowing the negatively charged neighboring acidic amino acids (15EDEEE19) to interact with the positively charged sequence surrounding H3K9me3 (8RKmeSTGGKAPRK18) [52]. Casein kinase II (CKII) has been identified as the responsible kinase to phosphorylate those residues [50,53,54]. CKII-mediated phosphorylation increases HP1α specificity for H3K9me3—contrary to non-phosphorylated HP1α, which binds with similar affinity to methylated and un-methylated H3K9 nucleosomes [49].

CKII also phosphorylates other HP1 sites. HP1β S89 and S175 have been reported as targets of CKII, but these phosphorylations do not seem to affect its chromatin binding ability [55]. However, in Drosophila, CKII-mediated S202 phosphorylation influences HP1 chromatin binding: mutation to a non-phosphorylable residue decreases its chromatin binding activity, and substitution to glutamate, which mimics phosphorylation, increased the binding activity. Interestingly, in other species’ HP1 homologs that do not have this site, the position is occupied by glutamate, which mimics constitutive phosphorylation [53,54]. Studies on fission yeast also pointed at CKII as the kinase that phosphorylates the HP1 homologue Swi6 at several residues; mutation of these sites to non-phosphorylable residues decreased transcriptional gene silencing, indicating that Swi6 phosphorylation is necessary for proper heterochromatin formation [56].

Other kinases have been identified to target HP1 sites. The serine/threonine-protein kinase Pim-1 [57] has been shown to directly interact with the chromoshadow domain of HP1α in yeast and mammalian cells: this interaction increases its binding to chromatin and negatively affects transcriptional repression [58]. During DNA damage, Homeodomain-interacting protein kinase 2 (HIPK2) phosphorylates the chromoshadow domain of HP1γ; this phosphorylation is important for HP1 interaction with H3K9me3 and might play a role in DNA damage repair by facilitating the accumulation of γH2A.X at damaged sites, which triggers the repair response [59].

### 3.2. Acetylation

HP1 proteins can also be acetylated in the hinge, chromo- and chromoshadow domains [44], although the specific acetyltransferases involved still remain unknown. Similar to what happens with histones, acetylation and methylation often occur at the same residue, which may suggest that these two PTMs act as a switch to alter HP1 binding [44]. However, to our knowledge, no studies to date have addressed how acetylation affects HP1 dynamics. This aspect would be an interesting avenue to develop further in order to evaluate if an Acetyl–Methyl switch exists for this chromatin reader, as it is for the histone code. It might be that a coordinated mechanism is in place to regulate both the histones and the reader in parallel in order to achieve a rapid and strong switch from activation to repression and vice versa.

### 3.3. Methylation

HP1 can also be methylated at several residues [16,44]; mono-methylation appears to be the most frequent type of methylation in HP1 [44].

The specific HP1 methyltransferases are still unknown. However, as LeRoy et al. (2009) pointed out, all three HP1 isoforms can directly interact with the methyltransferase Su(var)39H1 through its chromoshadow domain [60], so it might be that this methyltransferase is responsible for at least some of these methylations. Another possible methyltransferase could be SETDB1, as HP1 interacts with KAP-1, a protein that binds SETDB1 [61]. However, studies to corroborate these hypotheses in vitro or in vivo are yet to be performed.

### 3.4. Formylation

Lysine formylation has been reported in histones and other nuclear proteins and linked to chromatin function and regulation [62]. LeRoy et al. (2009) detected HP1 formylation not only in lysine residues, but also in serine and threonine residues; some of these sites were found in the sequence of the HP1 chromodomain that binds H3K9me3, suggesting that it might play a role in chromatin binding [44].

### 3.5. Citrullination

Another known modification of HP1 is peptidyl citrullination, where an arginine residue is converted to citrulline, a non-encoded amino acid, losing a positive charge and reducing the hydrogen bonding ability. Wiese and colleagues [43] found that the chromodomain of HP1γ is citrullinated by peptidylarginine deiminase 4 (PADI4) in mouse embryonic stem cells (mESCs) at R38 and R39, and that these modifications disrupt HP1γ-chromatin binding, thus supporting an open chromatin state in pluripotent mESCs. Upon differentiation, PADI4 expression levels drop and HP1γ citrullination is reduced, which stabilizes HP1γ-H3K9me3 binding and favors heterochromatin formation.

### 3.6. SUMOylation

Several studies have shown the implication of SUMO proteins in transcription repression regulation and maintenance of silenced heterochromatin [63]. HP1α can be SUMOylated at the hinge domain by SUMO1, which is crucial for its targeting to pericentric heterochromatin in mice [45]; the methyltransferase Su(var)39H1, but not Su(var)39H2, seems to enhance this SUMOylation [64]. Following HP1α deposition at the pericentric heterochromatin, HP1α SUMOylation is rapidly reversed by SENP7, necessary to stabilize and maintain its binding to chromatin [65].

In SUV39H1-depleted cells, SUMOylated HP1α was targeted to the chromatin but it could not be maintained [45], suggesting that SUMOylation of HP1 is required for de novo deposition of HP1 onto pericentric heterochromatin, while H3K9me3 is required for its maintenance. Understanding how this framework is regulated will provide new insights on constitutive heterochromatin dynamics and maintenance.

### 3.7. Ubiquitination

Ubiquitination is yet another important post-translational modification of HP1 and several groups have reported degradation of HP1 by several ubiquitin ligases.

During DNA damage, RAD6 has been linked to the ubiquitination of HP1 at K154, which promotes its degradation, forming an open chromatin state that will enable DNA repair [66].

This degradation system could be very important for the remodelling of chromatin that occurs during DNA damage. In fact, this targeted disruption could be coordinated with the eviction of histones at sites of DNA breaks, also proteasome-mediated, in order to facilitate the recruitment of repair factors [67].

The ubiquitin ligase RNF123 has also been shown to ubiquitinate and contribute to the degradation of HP1α and HP1β, but not HP1γ, upon lamin A/C mutations [68]. HECW2, another E3 ubiquitin ligase, has also been reported to target HP1α and HP1β for proteasomal degradation [69].

## 4. HP1 during Mitosis

When cells enter mitosis, chromatin is subjected to several changes that allow the formation of mitotic chromosomes for proper segregation of the genetic material. In addition to H3S10ph as a regulator of HP1-chromatin binding through mitosis, PTMs of HP1 itself also play a role in proper mitotic progression.

Early in this century, Mink et al. (2000) already reported that HP1α and HP1γ are hyperphosphorylated during mitosis [10]. Aurora A seems to be involved in phosphorylating HP1γS38 in G2/M phase, crucial for mitotic progression. HP1γ knockdown resulted in mitotic defects like multipolar spindles, centrosome defects and lagging chromosomes; this phenotype was rescued by the wild type HP1γ but not by the phospho-depleted mutant (S83A) [70], suggesting that phosphorylation at S38 is necessary for proper mitotic progression.

### HP1 Function at the Centromeres

Although the majority of HP1α is removed from the chromatin during mitosis, a small fraction is retained at the centromeres [71] where it is important for centromeric cohesion [72,73,74]. HP1α overexpression results in cohesion defects [72] while HP1α knockdown increases chromosome mis-segregation and micronuclei formation [73]: this indicates that proper levels of HP1α at the centromere are required for normal cell division. Figure 2 shows the main known functions of HP1 at the centromere.

How HP1α achieves this function at the centromeres has been studied by several groups, yet some aspects remain unclear. Although the role of H3K9me3 for HP1α-chromatin binding has been well studied, some evidence shows that the histone variant H2A.Z is also important for HP1α function at the centromeres [75,76,77]. How this is regulated still remains unknown, although it has been shown that HP1α can directly interact with H2A.Z [76] and that H2A.Z can substitute H3K9me3 and enhance HP1α binding to nucleosome arrays in vitro [78]. Newly synthetized H2A.Z is deposited at the pericentromeric heterochromatin during G1; this deposition seems to depend on the heterochromatin status, as Suv39H double-null cells present increased H2A.Z levels at pericentromeric heterochromatin [79], which could serve as a compensation for the loss of the H3K9me3 mark. Although more studies are needed to confirm this function of H2A.Z, it is clear that this histone variant plays an important role in HP1 regulation at the centromeres. Interestingly, H2A.Z exists as two distinct variants H2A.Z.1 and H2A.Z.2 which seems to have different functions in some context [80]; it would be therefore interesting to evaluate their contribution also in this context.

HP1α was also identified as one of the first known binding partners of the chromosomal passenger complex (CPC) [21], a crucial regulator of mitosis composed of the kinase protein Aurora B, the scaffold subunit inner centromere protein (INCENP) and two regulatory proteins Survivin and Borealin. The chromoshadow domain of HP1α has been shown to interact with INCENP and Borealin through a conserved chromoshadow domain-binding motif (PXVXL) present in both CPC subunits [21,22]. As these interactions require HP1α binding to the same motif in both proteins, they cannot occur at the same time. In fact, they appear to be time-regulated and have different functions: while the interaction with Borealin before mitotic entry is required for CPC recruitment and activation at the centromeres [22,81], HP1α–INCENP interaction during mitosis is important for HP1α localization at the centromere and proper chromosome segregation [82,83]. Disruption of the INCENP sequence responsible for HP1α binding affects chromosome segregation and sister-chromatids cohesion, a phenotype that can be rescued by tethering HP1α to the centromere [83]. HP1α–INCENP interaction increases centromeric localization of Haspin, which protects sister-chromatids cohesion [83,84]. A negative feedback loop might exist, as Haspin phosphorylates H3T3, which is important for CPC accumulation at mitotic centromeres [85,86,87,88,89].

HP1α–INCENP appears to be important for some Aurora B functions: while HP1α–INCENP disruption does not alter the levels of Aurora B-mediated H3S10 and CENP-AS7 phosphorylation [83], the phosphorylation of some Aurora B substrates at the kinetochores, namely Hec1 and Dsn1, is compromised [74]. Hec1 and Dsn1 phosphorylation is needed for kinetochore-microtubule attachment and proper chromosome segregation.

Shugoshin (Sgo1) is a protein that prevents premature cohesin dissociation [90]. Sgo1 also has a chromoshadow domain-interactive motif and is able to bind HP1α, but the disruption of this interaction does not seem to directly affect chromosome segregation [81]. In S. cerevisiae, HP1 homologue Swi6 was also seen to interact with cohesin; this interaction was shown to be necessary for cohesin targeting to the centromeres, where it is crucial to maintain sister-chromatids cohesion [91,92]. However, in human cells, cohesin accumulation at the centromeres was not affected by individual knockdown of the three HP1 isoforms [93]. The centromere structure is highly conserved between species [94] but the mentioned differences might indicate distinct HP1 regulatory mechanisms.

Although the role of HP1α in pericentric heterochromatin and chromosome segregation has been addressed, the implications of HP1β and HP1γ still remain controversial. Some studies report that all three HP1 isoforms co-immunoprecipitate with INCENP [74], while others show that INCENP is only able to bind HP1α and HP1γ [83,84]. Other studies observed HP1β associated with the centromeres in interphase, but not during mitosis [95]. Further studies would be needed to specifically show the functions of the distinct HP1 isoforms at the centromeres.

HP1α PTMs are also important for centromeric targeting; specifically, phosphorylation of the hinge region seems to regulate HP1α centromeric localization during mitosis. In early prophase, Aurora B phosphorylates HP1α at S92 [96]; this modification is important for the proper localization to the centromeres, in fact mutation of S92 to a non-phosphorylatable residue (S92A) decreased HP1α at the centromeres and led to anaphase bridges and micronuclei formation [97]. In addition, NDR1 kinase phosphorylates HP1α at S95, required for mitotic progression and SgoI loading into centromeres [98]. Inability to phosphorylate HP1 either by decreasing the responsible kinase or by introducing a non-phosphorylable HP1 mutant, leads to prometaphase arrest and several mitotic defects. During mitotic exit, S95 is dephosphorylated, coinciding with the release of SgoI from the centromeres, but the specific phosphatase remains unknown [99].

## 5. Conclusions

The role of epigenetics in human diseases has gained interest in recent years as our knowledge on epigenetic-regulated processes has increased. How chromatin is organized and structured has profound implications on both gene expression and genome stability and chromosome segregation, which can result in abnormal cell functions and lead to the origin of different diseases. Thus, studying the mechanisms involved in the regulation of chromatin architecture is crucial, not only to understand many aspects of cell biology, but also to address treatment of some diseases.

Protein regulation by post-translational modifications has been known for many years and it is now accepted that just studying the protein itself is not sufficient to fully understand its role. In this review, we summarized the recent literature that encompasses HP1 post-translational modifications and how they affect its chromatin binding, phase separation behavior and heterochromatin formation. Although several studies have identified the specific enzymes involved and the implications these modifications might have, Figure 1 clearly shows that most of them are still to be discovered. Any alterations in these proteins might influence heterochromatin formation and gene expression, potentially leading to several diseases.

## Figures and Tables

**Figure 1 cells-09-01460-f001:**
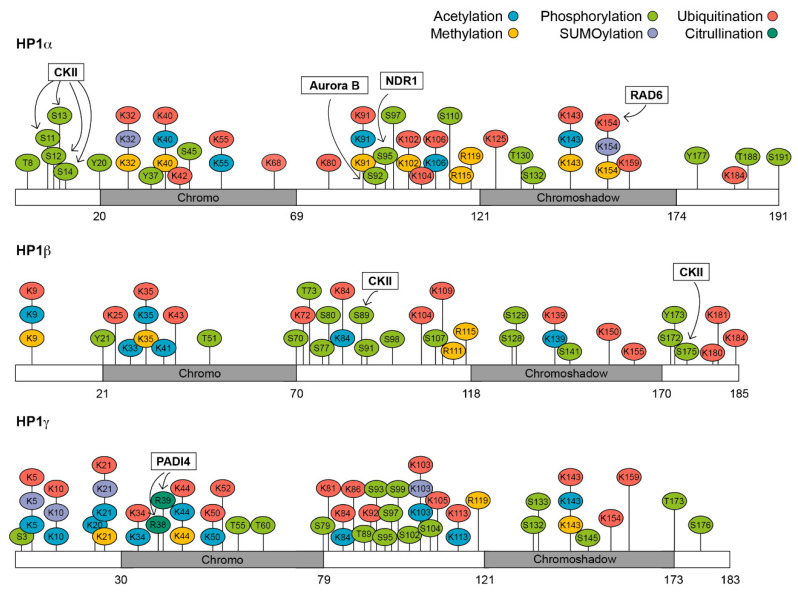
Heterochromatin Protein 1 (HP1) post-translational modifications. Acetylation (blue), methylation (yellow), phosphorylation (light green), SUMOylation (purple), ubiquitination (red) and citrullination (dark green) sites of HP1α, HP1β and HP1γ are shown. Black boxes highlight the known enzymes that catalyse specific modifications: Casein Kinase II (CKII); Peptidylarginine deiminase type 4 (PADI4); nuclear Dbf2-related kinase (NDR1); radiation protein 6 (RAD6). Grey boxes represent the chromo- and chromoshadow domains of HP1; the number below indicate the amino acids.

**Figure 2 cells-09-01460-f002:**
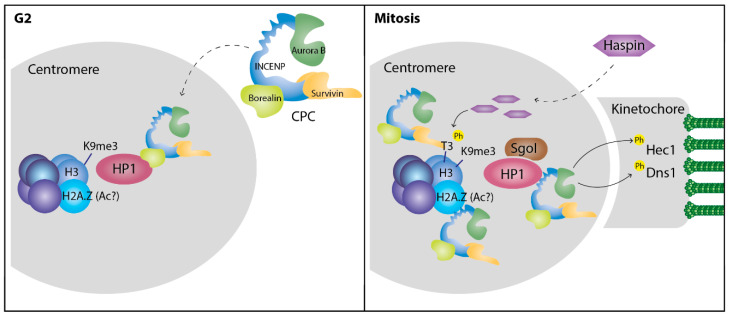
HP1 functions at the centromere. H3K9me3 and H2A.Z—probably the acetylated version (Ac?)—both play a role in maintaining HP1 to the centromeres. Before mitotic entry, HP1 binding to Borealin is important for the recruitment of the Chromosomal Passenger Complex (CPC) to the centromere. During mitosis, the CPC binds to HP1 through its subunit inner centromere protein (INCENP), which can also bind H2A.Z. This interplay might enhance or stabilize the CPC at the centromere to allow Aurora B-mediated phosphorylation (Ph) of several kinetochore proteins and proper microtubule attachment. HP1–INCENP interaction promotes Haspin accumulation at the centromere, where it phosphorylates H3T3, leading to further recruitment of the CPC via the interaction with Survivin. Shugoshin (SgoI) also binds HP1 at the centromere.

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
