# Peer review of "How HP1 Post-Translational Modifications Regulate Heterochromatin Formation and Maintenance"

_cells, 2020, doi:10.3390/cells9061460_

Round 1

Reviewer 1 Report

I enjoyed reading the proposed Review by Raquel Sales Gil and Paola Vagnarelli. This manuscript reviews a number of interesting articles on the HP1 post-transcriptional modification and their contribution on heterochromatin formation and maintenance.

I have some comments that might help improve the review:

  1. Line 25, it is mentioned that constitutive heterochromatin is found at pericentromeres and telomeres. Many reports have failed to detect high density of heterochromatic marks at telomeres but instead subtelomeres are the main heterochromatic core of those regions. Please look at this reference: Nucleic Acids Res. 2018 Mar 16; 46(5): 2347–2355. doi: 10.1093/nar/gky006

  1. In line 56-57, it is stated that the chromoshadow domain is involved in the interaction with other proteins. It would be worth to include at this stage few lines on the proteins HP1 interacts to.

  1. Please include the following reference :

Bannister, A., Zegerman, P., Partridge, J. et al. Selective recognition of methylated lysine 9 on histone H3 by the HP1 chromo domain. Nature 410, 120–124 (2001). https://doi.org/10.1038/35065138

  1. The use of alpha, beta and gamma should be checked all over the manuscript. In most cases only alpha is written e.g. line 41, 44, 102, 204….

  1. The use of pericentromeres and centromeres in the manuscript should be carefully checked. Although both structures share some structural features, pericentromeres are the core regions of constitutive heterochromatin, where most H3K9me3 concentrates, GC rich regions… on the contrary, centromeres have much less H3K9me3 and heterochromatic marks, are AT-rich etc… thus the two structures should not be mixed.

One of those examples is in the paragraph starting on line 285 where the authors started by saying ‘HP1a PTMs are also important for pericentromeric targeting’ but then all the paragraph is about centromeres.

Author Response

We thank the referee and we are glad he/she enjoyed reading about these aspects of HP1 biology.

These are our responses to the suggestions:

1_“Line 25, it is mentioned that constitutive heterochromatin is found at pericentromeres and telomeres. Many reports have failed to detect high density of heterochromatic marks at telomeres but instead subtelomeres are the main heterochromatic core of those regions. Please look at this reference: Nucleic Acids Res. 2018 Mar 16; 46(5): 2347–2355. doi: 10.1093/nar/gky006’

Line 25, We have changed telomeres to subtelomeres as the referee suggested and added the reference to the text [ref. 4].

2_“In line 56-57, it is stated that the chromoshadow domain is involved in the interaction with other proteins. It would be worth to include at this stage few lines on the proteins HP1 interacts to”

(now Line 57-58) we have added it briefly to maintain readibility: interaction with other proteins that contain the PXVXL motif, including TIF1, Chromatin Assembly Factor 1, INCENP and Borealin.  We added 5 references to support this claim [ref. 18 – 22]:

- Murzina, N.; Verreault, A.; Laue, E.; Stillman, B. Heterochromatin dynamics in mouse cells: interaction between chromatin assembly factor 1 and HP1 proteins. Mol Cell 1999, 4, 529-540.

- Le Douarin, B.; Nielsen, A.L.; Garnier, J.M.; Ichinose, H.; Jeanmougin, F.; Losson, R.; Chambon, P. A possible involvement of TIF1 alpha and TIF1 beta in the epigenetic control of transcription by nuclear receptors. EMBO J 1996, 15, 6701-6715.

- Thiru, A.; Nietlispach, D.; Mott, H.R.; Okuwaki, M.; Lyon, D.; Nielsen, P.R.; Hirshberg, M.; Verreault, A.; Murzina, N.V.; Laue, E.D. Structural basis of HP1/PXVXL motif peptide interactions and HP1 localisation to heterochromatin. EMBO J 2004, 23, 489-499.

- Ainsztein, A.M.; Kandels-Lewis, S.E.; Mackay, A.M.; Earnshaw, W.C. INCENP centromere and spindle targeting: identification of essential conserved motifs and involvement of heterochromatin protein HP1. J Cell Biol 1998, 143, 1763-1774.

- Liu, X.; Song, Z.; Huo, Y.; Zhang, J.; Zhu, T.; Wang, J.; Zhao, X.; Aikhionbare, F.; Zhang, J.; Duan, H. Chromatin protein HP1α interacts with the mitotic regulator borealin protein and specifies the centromere localization of the chromosomal passenger complex. J Biol Chem 2014, 289, 20638-20649.

3_“Please include the following reference :

Bannister, A., Zegerman, P., Partridge, J. et al. Selective recognition of methylated lysine 9 on histone H3 by the HP1 chromo domain. Nature 410, 120–124 (2001). https://doi.org/10.1038/35065138”

We added the reference in Lane 56, added [ref. 17]

4_”The use of alpha, beta and gamma should be checked all over the manuscript. In most cases only alpha is written e.g. line 41, 44, 102, 204….”

We are very sorry for this mistake, there was an issue in transferring the file into the journal template. We have changed them to the appropriate HP1 isoform in all the manuscript.

5_”The use of pericentromeres and centromeres in the manuscript should be carefully checked. Although both structures share some structural features, pericentromeres are the core regions of constitutive heterochromatin, where most H3K9me3 concentrates, GC rich regions… on the contrary, centromeres have much less H3K9me3 and heterochromatic marks, are AT-rich etc… thus the two structures should not be mixed.”

One of those examples is in the paragraph starting on line 285 where the authors started by saying ‘HP1a PTMs are also important for pericentromeric targeting’ but then all the paragraph is about centromeres.”

We agree with the referee that these compartments need to be correctly mentioned as they are not synonymous. We have carefully revised the terms and changed them where appropriate.

Reviewer 2 Report

In the current review, Sales-Giles & Vagnarelli summarize the recent literature on the heterochromatin protein 1 (HP1), with a particular focus on the post-translational modifications of HP1 and their role in heterochromatin formation. The review is well written and very informative. With its specific focus, it is highly relevant to the epigenetics community as it is covering an emerging field and I therefore recommend it for publication.

I have a few minor points that should be addressed:

- in the abstract is stated that HP1 binds to H3K9me2/3 (line 9) while is says H3K9me3 in the introduction (line 29)

- throughout the manuscript, HP1 beta and gamma were often mislabeled as HP1 alpha (see lines 41, 102, 124, 148, 204/205, 212)

- line 92: peak (without s)

- line 104: ‘discuss’ might be more appropriate than ‘analyze’

- Figure: the resolution of the figure is too low and the letters/numbers are too small to read

- Figure 1 legend: HP1a, b, g should be replace by alpha, beta, gamma

- line 126: What is meant with ‘extend and reduce the dynamics’?

- line 136: It is not clear to which HP1 homologs the authors refer?

- line 150: a reference is missing

- line 190/191: Do the authors suggest that H3K9me3 is not involved in recruitment, but only in maintenance of HP1?

- Figure 2: The histone tail is barely visible. Where would Haspin be placed in the scheme?

- line 244: ‘be’ might be replaced with ‘serve’

- line 288: it should read ‘non-phosphorylatable’

- line 299: since gene expression is barely discussed in the review, it might be replaced by ‘genome stability and chromosome segregation’

- line 310: as there are no diseases linked to HP1 discussed in this review, the last sentence appears very misplaced

Author Response

We thank the reviewer and we are glad to know that he/she considered the review well written, very informative and of highly relevant to the epigenetics community.

These are responses to the suggestions:

1_ “In the abstract is stated that HP1 binds to H3K9me2/3 (line 9) while is says H3K9me3 in the introduction (line 29)”

(now Line 29-30) we have changed H3K9me3 to H3K9me2/3 as in the abstract and according to the literature.

2_“Throughout the manuscript, HP1 beta and gamma were often mislabeled as HP1 alpha (see lines 41, 102, 124, 148, 204/205, 212)”

We are very sorry for this mistake, there was an issue transferring the file into the journal template. We have changed them to the appropriate HP1 isoform all around the manuscript.

 3_ Line 92: ‘peak (without s)”

(now Line 94) it has been corrected.

4_Line 104: “‘discuss’ might be more appropriate than ‘analyze’”

(now Line 106) we have changed it as suggested.

 5_”Figure: the resolution of the figure is too low, and the letters/numbers are too small to read”

We believe the low resolution is due to the insertion of the figure to the Word document. The original pdf files have higher resolution. Otherwise, please let us know and we will address the issue.

 6_”Figure 1 legend: HP1a, b, g should be replaced by alpha, beta, gamma’

These have been corrected.

7_line 126: “What is meant with ‘extend and reduce the dynamics’?”

(new Line 128)  We have changed the sentence to: ….they change the conformation of the N-terminal tail….

8_”line 136: It is not clear to which HP1 homologs the authors refer?”

(new Line 138), we have specified:  in other species’ HP1 homologs

9_”line 150: a reference is missing”

(new Line 152) we added the reference [ref. 44]

Ref.44: LeRoy, G.; Weston, J.T.; Zee, B.M.; Young, N.L.; Plazas-Mayorca, M.D.; Garcia, B.A. Heterochromatin protein 1 is extensively decorated with histone code-like post-translational modifications. Molecular & Cellular Proteomics 2009, 8, 2432-2442.

 10_Line 190/191: Do the authors suggest that H3K9me3 is not involved in recruitment, but only in maintenance of HP1?

(new Line 192-194). For clarification, we have mentioned the article suggesting that H3K9me3 is not involved in HP1 recruitment but rather it is necessary for its maintenance [ref. 45]. We also have rephrased the sentence in order to gain in clarity as follows:

In SUV39H1-depleted cells, SUMOylated HP1a was targeted to the chromatin but it could not be maintained [45], suggesting that SUMOylation of HP1 is required for de novo deposition of HP1 onto pericentric heterochromatin, while H3K9me3 is required for its maintenance.

Ref.45: Maison, C.; Bailly, D.; Roche, D.; de Oca, R.M.; Probst, A.V.; Vassias, I.; Dingli, F.; Lombard, B.; Loew, D.; Quivy, J. SUMOylation promotes de novo targeting of HP1α to pericentric heterochromatin. Nat Genet 2011, 43, 220

11_’Figure 2: The histone tail is barely visible. Where would Haspin be placed in the scheme?”

We have increased the size and changed the color of the histone tail.

We also agree with the referee that we should add Haspin in the picture in order to be complete. We have therefore modified the scheme to include this branch of the CPC targeting.  

We have also added the following to the figure legend: HP1-INCENP interaction promotes Haspin accumulation at the centromere, where it phosphorylates H3T3, leading to further recruitment of the CPC via the interaction with Survivin. 

12_”Line 244: ‘be’ might be replaced with ‘serve’”

(now Line 248) has been modified as suggested.

 13_”Line 288: it should read ‘non-phosphorylatable’”

(now Line 292) we corrected it.

14_”Line 299: since gene expression is barely discussed in the review, it might be replaced by ‘genome stability and chromosome segregation’”

(now Line 303) we added the referee’s suggestion, but we believe chromatin organization has profound implications on gene expression and should be left in the sentence, even if we have not addressed this topic in our manuscript.  

15_Line 310: “as there are no diseases linked to HP1 discussed in this review, the last sentence appears very misplaced”

The referee is correct to mention that we have not discussed any heterochromatin-related diseases in the manuscript. However, alteration on heterochromatin might lead to diseases and we wish to point that out. We have slightly rephrased the sentence as follows: (now Line 314) Any alterations on these proteins might influence heterochromatin formation and gene expression, potentially leading to several diseases.